# Co-Immobilization of Alcalase/Dispase for Production of Selenium-Enriched Peptide from *Cardamine violifolia*

**DOI:** 10.3390/foods13111753

**Published:** 2024-06-03

**Authors:** Shiyu Zhu, Yuheng Li, Xu Chen, Zhenzhou Zhu, Shuyi Li, Jingxin Song, Zhiqiang Zheng, Xin Cong, Shuiyuan Cheng

**Affiliations:** 1School of Modern Industry for Selenium Science and Engineering, Wuhan Polytechnic University, 36 Huanhu Middle Road, Wuhan 430048, China; zhushiyuu@126.com (S.Z.); jonathanlee0823@163.com (Y.L.); chxu@whpu.edu.cn (X.C.); lishuyisz@sina.com (S.L.); 13905189777@163.com (X.C.); s-y-cheng@sina.com (S.C.); 2Systems Engineering Institute, Beijing 100010, China; woshixiaoxin330@163.com

**Keywords:** selenium, peptide, co-immobilization, enzyme, *Cardamine violifolia*

## Abstract

Enzymatically derived selenium-enriched peptides from *Cardamine violifolia* (CV) can serve as valuable selenium supplements. However, the industrial application of free enzyme is impeded by its limited stability and reusability. Herein, this study explores the application of co-immobilized enzymes (Alcalase and Dispase) on amino resin for hydrolyzing CV proteins to produce selenium-enriched peptides. The successful enzyme immobilization was confirmed through scanning electron microscopy (SEM), energy dispersive X-ray (EDX), and Fourier-transform infrared spectroscopy (FTIR). Co-immobilized enzyme at a mass ratio of 5:1 (Alcalase/Dispase) exhibited the smallest pore size (7.065 nm) and highest activity (41 U/mg), resulting in a high degree of hydrolysis of CV protein (27.2%), which was obviously higher than the case of using free enzymes (20.7%) or immobilized Alcalase (25.8%). In addition, after a month of storage, the co-immobilized enzyme still retained a viability level of 41.93%, showing fairly good stability. Encouragingly, the selenium-enriched peptides from co-immobilized enzyme hydrolysis exhibited uniform distribution of selenium forms, complete amino acid fractions and homogeneous distribution of molecular weight, confirming the practicality of using co-immobilized enzymes for CV protein hydrolysis.

## 1. Introduction

Selenium (Se) is a vital micronutrient. However, on a global scale, Se distribution is uneven, with billions of people lacking sufficient dietary sources of Se. For example, research has reported that diseases such as cardiovascular disorders and tumors caused by Se deficiency affect populations in countries such as Russia, Serbia, and Belarus [1]. This deficiency can lead to the occurrence of related diseases, such as Keshan disease. Researchers have found that Se is not naturally synthesized within the human body and must be acquired from external sources. Therefore, the research and development of safe and effective Se supplements are key areas of current research [2,3,4]. In the development of Se-enriched food products, food safety has consistently been a paramount consideration. Emphasis should be placed on delivering functional foods that not only contribute to human health and enhance quality of life, but also ensure food safety throughout the consumption process. Currently, the Se supplements on the market are primarily plant-sourced organic Se, since supplementation of inorganic Se can easily lead to poisoning in the human body. In contrast, organic Se offers several advantages such as high bioavailability, strong biological activity, and enhanced safety, which is the ideal Se supplement for human consumption [5]. Organic Se in plants is generally present in the form of Se-enriched proteins, with selenocysteine (SeCys) and selenomethionine (SeMet) being the main forms [6]. SeCys is specifically encoded into the protein through specific genetic control, whereas SeMet is incorporated into the protein through random substitutions. Nasim et al. [7] confirmed that SeMet, proteins containing SeMet, and metabolites of SeMet together form a powerful triad of redox-active metabolites with a plethora of biological implications. Meanwhile, C. Liu et al. [8] proposed that the distinct reactivity of MeSeCys and SeMet towards different biologically relevant oxidants may explain their differing effectiveness as anticancer agents. Consequently, it is crucial to identify plant-derived sources of different Se forms.

*Cardamine violifolia* (CV) is a cruciferous plant with a widespread global distribution. Of notable mention is the CV grown in Enshi, Hubei Province, which is renowned for its remarkable Se enrichment capabilities [9]. In 2021, CV was approved by the National Health Commission of the People’s Republic of China for application as a food resource. As both a medicinal and edible plant, CV possesses rich nutritional value and research significance. Its potential in combating Se deficiency among populations worldwide is considerable [10]. Its application in food not only enriches people’s dietary options but also provides important raw material resources for the development of functional foods with health benefits. C. Du et al. [11] investigated the bioavailability of total Se and five Se compounds in CV leaves using an in vitro digestion combined with a Caco-2 cell monolayer model. They found that the bioavailability of total Se was 49.11%, with MeSeCys and SeMet having higher bioavailability than SeCys_2_. It has been found that Se in CV is predominantly found in Se-enriched proteins, which have a higher content of SeCys compared to SeMet [12], thus it is crucial to explore the Se forms of the different active constituents in CV. Compared to proteins, peptides possess advantages such as easy absorption and high activity. Additionally, antioxidant peptides derived from food sources typically do not exhibit activity in their natural structure. However, once hydrolyzed from macromolecules, they can exert biological activity, including scavenging free radicals, resisting oxidative stress, and other health-related physiological functions [13]. R. Ling et al. [14] prepared Se-enriched peptides by enzymatic hydrolysis of CV proteins using Alcalase and Dispase, and found that Se-enriched peptides increased mRNA expression of GPX1 and NGFR in ethanol-treated L-02 hepatocytes, confirming their protective effect against ethanol-induced liver injury. Q. Zhao et al. [15] prepared egg yolk Se-enriched peptides using Alcalase and Dispase hydrolysis combined with shear pretreatment, which exhibited excellent immunoassays, including immune organ index, serum levels of immune factors, splenic histopathological changes, and hepatic Se content. Notably, the synergistic effect of Se and peptides not only promotes Se absorption in the human body [16], but also enhances the biological activity of the peptides themselves [17]. For instance, B. Chen et al. [18] found that Se-enriched peptides enhanced the capacity for scavenging free radicals in vitro by blocking the Keap1-Nrf2 interaction and subsequent activation of the antioxidant stress response. While the research on the Se-enriched protein/peptides of CV has predominantly focused on their functional and morphological analysis, there have been fewer studies on the enzymatic hydrolysis of CV protein to produce peptides with a uniform distribution of Se forms.

The utilization of protease enzymatic digestion for peptide preparation has gained growing focus due to its safety and controllability, but then comes the drawback that the free enzyme is unstable and not recyclable [19]. Immobilized enzymes, which are widely used in the food industry, have more stable enzyme properties and can be recycled after the reaction is completed [20], showing great potential for Se-enriched peptide preparation. On the other hand, researchers have reported that the selection of enzyme type and control of enzymatic conditions enable the preparation of active peptides with specific physiological functions. For instance, Anwar et al. [21] suggested utilizing Alcalase and Papain to hydrolyze Chinese sturgeon protein, resulting in fractions and purified peptides that could serve as botanical antioxidants in pharmaceuticals and food items. In contrast to the use of a single enzyme, Tacias-Pascacio et al. [22] discovered that the biological activity of protein hydrolysates was further enhanced when Alcalase was used in combination with other proteases. X. Zhang et al. [23] demonstrated that the activity of Neutrase (also called Dispase) hydrolysis unveiled particular amino acid residues susceptible to Alcalase targeting, thereby producing additional antioxidant fragments. Furthermore, Y. Wang et al. [24] showed that immobilized dual enzyme exhibited a higher zein DH (65.8%) compared to free enzyme (49.3%) and single enzyme immobilized in calcium alginate beads (45.5%). However, there remains a scarcity of research on the preparation of distinct Se-enriched peptides from the hydrolysis of CV proteins by co-immobilized enzyme systems.

Herein, the research aims to investigate the feasibility of co-immobilized enzyme hydrolysis for producing Se-enriched peptides from CV protein. Co-immobilization of two enzymes (Alcalase and Dispase) was used to study the hydrolysis of CV proteins, which ultimately produced Se-enriched peptides with uniform Se forms and amino acid distribution. Alcalase and Dispase were chosen for their distinct recognition sites, which are expected to have a synergistic effect during enzyme immobilization, thus enhancing the overall co-immobilized enzyme activity. The resin was chosen as a carrier because its abundant amino groups could provide suitable immobilization sites for various enzymes, thereby enhancing the efficiency and activity of enzyme immobilization. In this study, the structure and activity of different co-immobilized enzymes were explored by adjusting the ratio of Alcalase and Dispase. On this basis, the effects of such changes on the amino acid composition and Se morphology distribution of the hydrolysis products were analyzed in detail. The results of this research are expected to provide more options and ideas in the field of hydrolysis of plant proteins by co-immobilized enzymes.

## 2. Materials and Methods

### 2.1. Materials

Alcalase (2 × 10^6^ U/g, BRENDA: EC3.4.21.62) from *Bacillus licheniformis* and Dispase (1 × 10^6^ U/g, BRENDA: EC3.4.24.28) from *Bacillus subtilis* were supplied by Solarbio Co., Ltd. (Beijing, China). Resin (LXTE-704), which was provided by Sunresin New Materials Co., Ltd. (Xi’an, China), possesses the following characteristics: particle size range of 0.1–0.3 mm, moisture content of 64.56%, and tertiary amine functional group capacity of 0.87 mmol/g. Bovine serum albumin (BSA, 96%) and tyrosine (98%) were obtained from Sigma-Aldrich. Hydrochloric acid (HCl), anhydrous ethanol (99.5%), trichloroacetic acid (TCA, 10%), sodium chloride (NaCl), sodium dihydrogen phosphate (NaH_2_PO_4_), disodium hydrogen phosphate (Na_2_HPO_4_), sodium hydroxide (NaOH), acetonitrile, citric acid, sodium hexanesulfonate, methanol, sodium acetate, and glutaraldehyde (GA) were purchased from Sinopharm Chemical Reagent Co., Ltd. (Shanghai, China). All the reagents utilized were of analytical or HPLC grade.

CV was acquired from Enshi Se-Run Material Engineering Technology Co., Ltd. (Enshi, China) [25]. The raw material was crushed into powder, filtered through an 80-mesh filter and maintained in a −18 °C refrigerator until use.

### 2.2. Extraction of Se-Enriched Protein from CV

Considering the highest yield of CV protein in alkaline solution, the CV protein was prepared using an alkaline extraction and acid precipitation method [26]. Specifically, 4 g of crushed CV powder was weighed and the alkali solution (0.1 M of NaOH solution) was subsequently added using a solid–liquid proportion of 1:40 g/mL [27]. After stirring for 8 h at 50 °C, the supernatant was obtained through centrifugation at 4000 rpm for 15 min, following which the pH was subsequently adjusted to the protein’s isoelectric point of 3.5.

During this process, the protein content in the supernatant and the raw material was determined using the Kjeldahl method [28]. The extraction yield of CV protein was calculated using the following formula:(1)$$E\mathrm{xtraction}\;\mathrm{yield}\;\mathrm{of}\;\mathrm{CV}\;\mathrm{protein}\;(\%)=\frac{M_{0}}{M_{1}}\;\times \;100\%$$
where the protein contents (g/100 g) in the supernatant and raw material are represented by M_0_ and M_1_, respectively.

Next, the powder of CV protein was obtained by overnight incubation at 4 °C, centrifuging at 4000 rpm for 15 min, in which the precipitate was gathered and subjected to dialysis for 48 h after adjusting the pH to 7, and finally freeze-dried.

### 2.3. Optimizing Enzymatic Hydrolysis of CV Protein by Free Alcalase

Considering that most of the CV protein mentioned above is alkali soluble, Alcalase was first used in the enzymatic hydrolysis experiment to explore the optimal enzymatic hydrolysis conditions. Here, the influence of solid-to-liquid ratio (S/L), enzyme-to-substrate ratio (E/S), and pH on enzymatic hydrolysis were investigated, with the main objective being the determination of the degree of hydrolysis. In this study, CV protein was dissolved in deionized water, and the pH of the reaction system was adjusted using 0.1 M HCl and NaOH. In detail, the ratio of S/L was adjusted to 1%, 2%, 3%, 4%, and 5% at an enzymatic hydrolysis temperature of 50 °C, time of 2 h, pH of 9.5, and E/S ratio of 20% to assess the impact of S/L ratio on enzymatic hydrolysis. The E/S ratio was regulated to 10%, 15%, 20%, 25%, and 30% at a temperature of 50 °C, time of 2 h, pH of 9.5, and S/L ratio of 3% to identify the influence of E/S ratio during enzymatic hydrolysis. The pH was realigned to 8.5, 9.5, 10.5, 11.5, and 12.5 at a temperature of 50 °C, time of 2 h, S/L ratio of 3%, and E/S ratio of 20% to identify the influence of pH on enzymatic hydrolysis.

### 2.4. Preparation of Co-Immobilized Enzyme

The screened resin with amino groups was selected. The resin was first washed 3 times with phosphate buffer (0.1 M, pH = 9.5), dried, and stored in a 4 °C refrigerator. Note that the resin should not be placed in a freeze-drying refrigerator or subjected to magnetic stirring for cleaning, as this may cause the resin to break [29].

#### 2.4.1. Determination of the Optimal Alcalase/Resin Ratio

The resin was activated by weighing a certain mass and subjecting it to agitation on a vibratory shaker with 8% GA in distilled water at 20 °C and 120 rpm for 1 h to ensure the uniformity of the activation degree. The activated resin was repeatedly rinsed with distilled water to remove residual GA from the surface, which was applied directly to the immobilization of the enzyme in order to avoid oxidation of the active groups on the resin surface.

To investigate the impact of Alcalase/resin ratios (m/m) (0.3, 0.5, 0.7, 0.9, 1.1) on enzyme immobilization, activated resin and Alcalase were mixed in a 25 mL triangular flask, with Alcalase dissolving in 20 mL of 0.1 M phosphate buffer at pH 9.5. The resulting suspension was stirred in a shaker at 20 °C and 120 rpm for 4 h, facilitating the enzyme’s covalent immobilization onto the activated resin. During the immobilization process, 0.1 M NaCl was introduced to maintain the enzyme’s conformation. Subsequently, the filtered resin underwent multiple rinses in a buffer solution until no free Alcalase was detectable in the washing solution, yielding the immobilized Alcalase. Finally, the immobilized Alcalase was dried in a vacuum desiccator at room temperature and then stored in a refrigerator at 4 °C.

The Bradford method was employed to ascertain the enzyme loading [30], employing the subsequent formula for determination:(2)$$\mathrm{Enzyme}\;\mathrm{loading}\;\mathrm{rate}\;(\%)=\frac{C_{0}\times V_{0}-C_{W}\times V_{W}}{C_{0}\times V_{0}}\;\times \;100\%$$
where the protein concentrations (mg/mL) in the initial solution and washing solution are represented by C_0_ and C_W_, respectively, and the volumes (mL) of the initial solution and washing solution are denoted by V_0_ and V_W_, respectively.

Herein, BSA (2 mL, 5 mg/mL) was used as the substrate to measure enzyme activity [31]. The performance of immobilized enzymes, referred to as matrix activity, was documented per milligram of matrix for easy comparison. BSA was mixed with enzyme in phosphate buffer (0.1 M, pH = 9.5) and stirred at 40 °C for 10 min. Subsequently, to cease the hydrolysis reaction, an excess of TCA (10%) was added to precipitate any undigested BSA. Following the reaction, the mixture underwent cooling to room temperature, with subsequent measurement of its absorbance at 275 nm [32,33].

#### 2.4.2. Determination of the Optimal Ratio of Two Proteases

Considering the variation of pH during enzymatic hydrolysis and the ability of CV protein to achieve the highest degree of hydrolysis (DH) after dual enzyme hydrolysis, Dispase was introduced. The specific immobilization method is the same as described in Section 2.4.1, and Alcalase was replaced by the mixture of Alcalase and Dispase. The optimal mass ratio of the two proteases (Alcalase/Dispase = 11:1–1:1) was screened with DH and enzyme activity as the main criteria, and the total enzyme amount was determined based on the optimal Alcalase/resin ratio in Section 2.4.1. The prepared CV protein powder was solubilized in deionized water with an S/L ratio of 3%, and the enzyme was added at the E/S ratio of 20% after adjusting the pH to 9.5 at 50 °C for 2 h. The mixture was boiled for 10 min to inactivate the enzyme, and the supernatant was centrifuged for 10 min to determine the effect of simultaneous extraction of both enzymes (Alcalase and Dispase = 11:1–1:1) on enzymatic hydrolysis. The pH-stat method was employed to determine the DH of CV protein, as described in a prior investigation [25].

### 2.5. Characterization

The pore size distribution, pore volume, and porosity of the resin, activated resin, immobilized Alcalase, and co-immobilized enzyme were analyzed with a specific surface pore size analyzer and a static nitrogen adsorption instrument (JW-BK112, Beijing JWGB SCI. & TECH. Co., Ltd., Beijing, China). The specific surface area was calculated and the isothermal adsorption and desorption curves were plotted using the Brunauer–Emmett–Teller (BET) formula. The pore diameter was determined using the Barrett–Joyner–Halenda (BJH) method. The form and energy dispersive X-ray (EDX) of activated resin, immobilized Alcalase, and co-immobilized enzyme samples were characterized by COXEM (EM30AX, OPTON, Beijing, China), and different areas of the samples were randomly selected for elemental analysis. Fourier-transform infrared (FTIR) spectra were utilized for the analysis of the secondary structure within a wavenumber range of 4000–400 cm^−1^ (Nexus 670, ThermoFisher Scientific, Madison, WI, USA).

### 2.6. Enzymatic Properties Assay of Free and Co-Immobilized Enzyme

#### 2.6.1. Examination of the Stability

To assess the thermal stability of the prepared co-immobilized enzyme, they were immersed in phosphate buffer (0.1 M, pH = 9.5) at 50 °C for different times (0–150 min) and then cooled to room temperature. The enzyme activity was determined by centrifugation at 8000 rpm/min for 10 min immediately after enzymatic digestion, and the supernatant was removed for enzyme activity determination. To assess the storage stability of free and co-immobilized enzymes, they were preserved in phosphate buffer at 4 °C. The enzyme activity was then tested under the optimal conditions at regular intervals of 5 days, up to a duration of 30 days. In terms of reusability, the co-immobilized enzyme was separated by centrifugation and underwent multiple washes after each hydrolysis process. The collected co-immobilized enzyme was subsequently employed for subsequent cycles of hydrolysis under identical conditions. The operational stability of the co-immobilized enzyme, as explained earlier, was evaluated by calculating the enzyme activity ratio in relation to the initial activity for each cycle, which was considered to be 100%.

#### 2.6.2. Kinetic Parameters of Enzyme

DH is proportional to the volume of NaOH consumed, so the reaction rate can be expressed in terms of the volume B of sodium hydroxide consumed per unit time [34]. A coordinate graph is created with time as the axis, and its initial slope dB/dt is the initial rate of the enzymatic reaction. Taking the amount of peptide bond broken per unit time, A, as the reaction rate, the initial slope is calculated as follows:dA/dt = d[B × N × (1/α)]/dt = N × (1/α) dB/dt(3)
where B, N, and α are the alkaline consumption (mL), the molar concentration of sodium hydroxide (mol/L), and the average dissociation degree of protein amino groups, respectively.

Since N × (1/α) is a constant value, the volume of sodium hydroxide consumed is used to represent the amount of peptide bond reacted. In this way, the initial reaction rate corresponding to the substrate concentration can be measured, and Km is obtained according to the Lineweaver–Burk double inverse equation:1/V = Km/Vmax × 1/[S] × 1/Vmax(4)

### 2.7. Analysis of Hydrolysis Products

#### 2.7.1. Measurement of Amino Acid Fraction

A 2 mL enzymatic solution after the end of hydrolysis was directly treated with HCl (3 mL, 6 M) at 110 °C for 24 h. Under nitrogen protection, the sample was dried in a water bath at 80 °C. Following this, it underwent multiple rinses with derivatization buffer solution before being transferred to a 25 mL volumetric flask. Subsequently, it was filtered through a 0.45 μm microporous membrane and set aside. The Elite-AAK amino acid analysis system employed 2,4-dinitrofluorobenzene (DNFB) as a pre-column derivatization reagent. Under specific conditions, amino acids react with DNFB to form dinitrophenylphenylalanine derivatives (DNP-AA) with UV absorbance, which are subsequently separated and detected. The Elite-AAK amino acid analysis column was 4.6 mm × 250 mm, 5 μm; the mobile phase A was acetonitrile (99%), and the mobile phase B was sodium acetate (0.05 M, pH = 5.28) at a flow rate of 1.1 mL/min, employing gradient elution. The detection was performed at 360 nm, while maintaining the column temperature at 24 °C.

#### 2.7.2. Determination of Se Content

First, we precipitated the unreacted protein from the enzymatic solution with an excess of TCA. Total Se and inorganic Se content were calculated by atomic fluorescence spectrometry according to the method of Shiyu Zhu et al. [35]. The content of organic Se was determined by subtracting the inorganic Se from the total Se.

After enzymatic hydrolysis, the separation of Se forms of the samples was achieved within 10 min through the usage of a C_18_ column (4.6 mm × 250 mm, 5 μm) at 25 °C with a flow rate of 1.0 mL/min. During this process, the mobile phase consisted of 10 mM citric acid and 5 mM sodium hexanesulfonate (pH = 4), with the addition of 1% methanol.

#### 2.7.3. Identification of Se-Enriched Peptide Sequences

The enzyme digestion products were lyophilized and desalted, and then separated by high-performance liquid chromatography (HPLC). The chromatographic conditions were Zorbax 300SB-C18 column (0.15 mm × 150 mm); Solvent A: 0.1% formic acid; Solvent B: 0.1% formic acid-acetonitrile (84% for acetonitrile); the gradient elution conditions were 0–50 min, A (96–50%), B (4–50%); 50–54 min, A (50–0%), B (50–100%); 54–60 min, A (0%), B (100%); the injection volume was 30 μL; the injection concentration was 1 mg/mL; the flow rate was 1.0 mL/min; the detection wavelength was 230 nm; the column temperature was 25 °C.

Se-enriched peptides were determined by Q Exactive Orbitrap Mass Spectrometers (Thermo Fisher Scientific, San Jose, CA, USA). The analysis time was 60 min. The detection mode was positive ion. The mass/charge ratios of peptides and peptide fragments were determined using the following approach: 10 fragment profiles were acquired after each full scan (MS2 scan).

The mass spectrometry test raw files were searched in the corresponding databases using the software MaxQuant 1.5.5.1, and the protein identification and quantitative analysis yielded the final results.

### 2.8. Statistical Analysis

Our experiments were conducted with three repetitions, and the data were analyzed using SPSS 19.0. Variations among experiments conducted on individual levels were managed through analysis of variance (ANOVA) and evaluated with Duncan’s range test at a significance level of 5%. Presenting the results includes expressing them as mean values ± standard. Different letters in the same test indicate significant differences (*p* < 0.05).

## 3. Results and Discussion

### 3.1. Enzyme Immobilization

#### 3.1.1. Single-Factor Experiment of Free Alcalase

The results based on the important factors affecting DH indicated that an increase in the S/L ratio from 1% to 3% led to an increase in DH (Figure 1). However, excessive S/L ratios resulted in a slight decrease in DH due to a type of uncompetitive inhibition known as high-substrate inhibition, which led to a decrease in the hydrolysis rate [36]. Moderate increases in the E/S ratio resulted in a gradual enhancement of DH. This was attributed to the beneficial effect of increasing the enzyme quantity, facilitating improved interaction between the enzyme and substrate [37]. However, once the enzyme concentration exceeded the available number of cleavable bonds, further increasing the enzyme concentration did not increase the degree of hydrolysis. Moreover, within the pH range increasing from 7.5 to 9.5, there was an increase in DH. This was likely attributed to the facilitated recognition and hydrolysis of alkali-extracted proteins by enzymes under alkaline conditions. But further elevation of pH may lead to a reduction in enzyme activity and could potentially impact protein structure, resulting in a decrease in DH [38].

#### 3.1.2. Determination of the Optimal Alcalase/Resin and Alcalase/Dispase Ratio

The CV protein was extracted through alkaline extraction and acid precipitation (with an extraction yield of 50.38 ± 0.64%), resulting in most of the proteins being soluble in alkali. As the hydrolysis reaction progressed, the generation of carboxyl-containing amino acids released protons into the solution, leading to a decrease in pH, transitioning the system from alkaline to neutral. To optimize the hydrolysis of CV protein under the most favorable conditions, Alcalase was initially chosen for immobilization to determine the optimal Alcalase/resin ratio. Subsequently, Dispase was introduced to investigate the impact of the Alcalase/Dispase combination on the structure and activity of co-immobilized enzymes. It was found that the highest DH and enzyme loading (405.56 ± 0.46 mg/g) were obtained with an Alcalase/resin ratio of 0.9, which was 10 times higher than the immobilized enzyme loading (37.7 mg/g_galV_) prepared by Tan et al. [39]. The enzyme loading significantly decreased with additional Alcalase, as excessive Alcalase would lead to a spatial blockage between enzyme molecules [40,41]. Therefore, the Alcalase/resin ratio of 0.9 was chosen for subsequent experiments. After Dispase addition, the enzyme loading increased significantly from 45.06% to 77.72%, and the enzyme activity firstly experienced a substantial boost (Figure 2d) with the increasing amount of Dispase peaking at a ratio of 5:1 (Alcalase/Dispase) for the two enzymes.

### 3.2. N_2_ Adsorption–Desorption and Pore Size Distribution of Co-Immobilized Enzyme

Figure 3 displays the BET N_2_ isothermal adsorption–desorption curves for the resin, activated resin, and the co-immobilized enzyme, and they were all typical IV-type isothermal adsorption–desorption curves for mesoporous materials [42], where the relative pressure corresponding to the step zone reflects the size of the mesoporous material diameter. Notably, compared to other co-immobilized enzymes, when the ratio of the two enzymes was 5:1, the isothermal adsorption–desorption curve exhibited a stepped adsorption pattern at relatively low relative pressures (the red arrow in Figure 3) and maximum activity, suggesting that changes in adsorption at lower relative pressures may be attributed to specific filling of microporous structures and the formation of adsorption sites [43]. Importantly, this implied a mutual enhancement between the two enzymes at this ratio, leading to a synergistic effect that enhanced the performance of the co-immobilized enzyme [44]. Additionally, the specific surface area and pore size were measured. Table 1 indicates that the purchased resin material possessed a large specific surface area (113.68 m^2^·g^−1^), confirming its potential as a carrier for enzyme immobilization. After activation of the resin with GA, new coatings were formed, leading to a decrease in the specific surface area (50.261 m^2^·g^−1^). However, after enzyme co-immobilization, the specific surface area increased again. This could be attributed to enzyme molecules adsorbing onto the pores and surface of the resin, forming new irregular voids, especially within the mesopore range, during the co-immobilization process [45]. Meanwhile, when the ratio of the two enzymes was 5:1, the smallest pore size of the immobilized enzyme was measured at 7.065 nm, indicating effective adsorption of enzyme molecules at this point [46].

### 3.3. Characterization of Co-Immobilized Enzyme

The morphological characteristics of the activated resin, immobilized Alcalase, and co-immobilized enzyme_5:1_ were observed by scanning electron microscopy (SEM). As shown in Figure 4a, the activated resins exhibited a spherical shape with particles of uniform size and smooth surfaces. Following the immobilization of Alcalase, there was a slight increase in the particle size of the microspheres (from 205.99 μm to 211.39 μm), and the surface became rougher (Figure 4b), indicating the adsorption of enzyme molecules [47]. Upon the addition of Dispase and adjustment to optimal conditions (Alcalase/Dispase = 5:1), the particle size of the microspheres further increased, illustrating that the enzyme molecules were more easily immobilized on the activated resin after the introduction of Dispase [48], which is consistent with our previous results on pore size distribution of the co-immobilized enzyme. Moreover, the activated resin consisted of methyl acrylate and served as a carrier for enzyme immobilization. The FTIR spectra of the activated resin before and after immobilization are shown in Figure 5. The activated resin exhibited characteristic peaks of the carbonyl group (C=O) vibrations of the ester at 1730 cm^−1^ and methyl vibrations at 2956 cm^−1^, while the characteristic peak of the amide bond (R−CONH) was located near 1655 cm^−1^ [49]. In addition, the peaks appearing at 3200–3600 cm^−1^ are characteristic peaks of overlapping O−H and N−H [50]. After introducing enzymes into the activated resin, the transmittance of these main characteristic functional groups significantly decreased. Next, elemental analysis was performed, and the results are shown in Figure 6, revealing a notable increase in the proportion of S element after enzyme immobilization. This confirmed the successful immobilization of the enzyme, which is rich in S element and is known as a sulfhydryl protease.

### 3.4. Examination of the Performance of Co-Immobilized Enzyme

During the initial phase of storage at 50 °C (0–120 min), the activity recovery of the free enzyme was higher compared to the prepared co-immobilized enzyme (Figure 7). Interestingly, a reversal occurred after 150 min of storage, where the activity recovery of free enzyme was lower than that of the co-immobilized enzyme. This phenomenon may be attributed to the stability and structure of the enzyme on the resin [51]. Investigating the reusability of co-immobilized enzymes revealed that after four and seven cycles, they retained 48.56% and 20.49% of their activity, respectively. The observed activity loss may be attributed to enzyme molecules leaching from the resin during the washing stage [52]. The prepared co-immobilized enzyme_5:1_ maintained 41.93% of its activity after one month of storage. Among them, the loss of enzyme activity of the co-immobilized enzyme_5:1_ was higher in the first ten days, probably due to the natural shedding of loosely bound enzyme molecules during storage. The activity of the co-immobilized enzyme_5:1_ decreased more slowly compared to the free enzyme over the following 20 days, suggesting its stability and increased resistance to storage [53].

The kinetic parameters of the co-immobilized enzyme were then investigated, and it was revealed from Figure 8 that the K_m_ and V_max_ values of the co-immobilized enzyme_5:1_ were larger than those of the free enzyme, implying a decreased binding of the enzyme to the substrate. This could be attributed to the random rearrangement of the active site of the enzyme molecule during the immobilization process and the size of the resin pore restricting sufficient contact between the macromolecular substrate and the catalytic active site of the enzyme [54,55].

### 3.5. Determination of Enzymatic Digestion Products

Peptides hydrolyzed by different proteases exhibited distinct amino acid composition and sequence profiles. Alcalase, a non-specific endopeptidase, preferentially recognized Glu, Leu, Lys, Tyr, and Met amino acid residues at its enzyme recognition site. Its activity was found to be broad and peaked under alkaline conditions. In contrast, Dispase was also an endopeptidase, but it specifically targeted the N-terminal amino acid residues of Leu, Phe, and Tyr [23]. It typically operated within a narrow pH range (pH 5 to 8) and exhibited relatively low thermotolerance [56]. The relative ratios of various amino acids are depicted in Figure 9, clearly indicating significant changes in the ratio of certain amino acids (Asp, Glu, Ser) with variations in the proportions of the two proteases. In particular, it was observed that increasing the amount of neutral protease led to a subsequent increase in the content of Tyr, an amino acid that was initially undetected (Table 2). This phenomenon was likely a result of Dispase effectively hydrolyzing the residues targeted by Alcalase, allowing for further hydrolysis by Alcalase and subsequent production of more amino acid fragments. Moreover, regardless of the enzymatic digestion method, the CV protein enzymatic digestion product exhibited a higher content of sweet amino acids compared to bitter amino acids, suggesting a favorable overall taste profile. Based on the results above, the CV protein hydrolysate possessed a well-balanced amino acid composition, suggesting its suitability as a dietary resource for human consumption [57,58].

The hydrolysates at different Alcalase/Dispase ratios was treated with TCA to remove protein precipitation, and the total and organic Se content of the supernatant peptides were tested. The raw material of CV was found to have a high organic Se content of up to 1817.71 mg/kg (Table 3), primarily in the form of Secys_2_ (up to 88.88%). Moreover, the organic Se content of the alkali-extracted protein was 981.19 mg/kg, with 72.75% being SeMet. Interestingly, hydrolysis using the co-immobilized enzyme_5:1_ led to a homogeneous distribution of Se form and molecular weight in the CV peptides (Figure 10 and Table 4). The resulting Se-enriched peptide product contained 93.56% organic Se, highlighting its high biological activity and easy absorption by the human body, making it an ideal Se supplement due to its uniform distribution of Se form.

### 3.6. Comparison of the Efficiency of Different Immobilized Enzymes

Table 5 compared the efficiency and applications of different immobilized enzymes. It was found that, compared to other immobilized enzymes, the immobilized enzymes prepared with activated resin as a carrier in this study not only had a high enzyme loading and immobilization yield but also exhibited higher enzyme activity. This was primarily attributed to the resin carrier having an appropriate porous structure, preventing the clogging of substrate diffusion channels after enzyme molecule adsorption. Additionally, in comparison to our previous research [35], the DH of CV protein significantly increased (from 23.2% to 27.2%). Compared to other studies, the co-immobilized enzyme achieved a higher DH within a short period of time, which was attributed to the enrichment of substrates by the porous structure of the resin, and showed the promising application of the co-immobilized enzyme prepared in this study. However, it is important to note that although these co-immobilized proteases showed high hydrolysis under hydrolysis of plant proteins, the limitations of the immobilization process that might lead to a decrease in enzyme activity need to be addressed by further studies.

## 4. Conclusions

In summary, this study explored the viability of co-immobilized enzymatic hydrolysis for converting CV protein into Se-enriched peptides. The successful immobilization of Alcalase and Dispase on amino resin was confirmed through SEM, EDX, and FTIR. The co-immobilized enzyme exhibited the smallest pore size (7.065 nm) and highest viability (41 U/mg) with a 5:1 (m/m) ratio of Alcalase/Dispase. Compared to free enzymes and enzymes fixed only with Alcalase, the co-immobilized enzyme significantly increased the degree of hydrolysis of CV protein to 27.2%. After a month of storage, the co-immobilized enzyme still showed a viability level of 41.93%. In addition, the organic Se in CV and CV protein mainly existed in the form of Secys_2_ and SeMet, respectively, while the Se-enriched peptides from co-immobilized enzyme hydrolysis of CV protein exhibited a uniform distribution of Se forms, as well as complete amino acid fractions and a homogeneous distribution of molecular weight. This study provided novel insights into the field of enzymatic immobilization for plant protein hydrolysis. Future research needs to further use co-immobilized enzymes with high enzyme activity to prepare Se-enriched antioxidant peptides, investigate their safety, and study their biological effects in cells.

## Figures and Tables

**Figure 1 foods-13-01753-f001:**
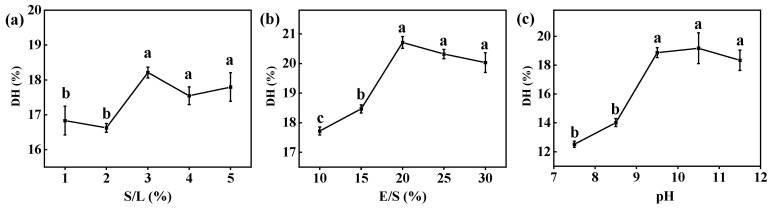
Effect of (**a**) S/L, (**b**) E/S, and (**c**) pH on DH during hydrolysis of CV protein by free Alcalase. Reaction conditions: (**a**) temperature = 50 °C, time = 2 h, pH = 9.5 (deionized water), and E/S = 20%. (**b**) temperature = 50 °C, time = 2 h, pH = 9.5 (deionized water), and S/L = 3%. (**c**) temperature = 50 °C, time = 2 h, S/L = 3%, and E/S = 20%. Different letters indicate significant differences (*p* < 0.05) in the same column.

**Figure 2 foods-13-01753-f002:**
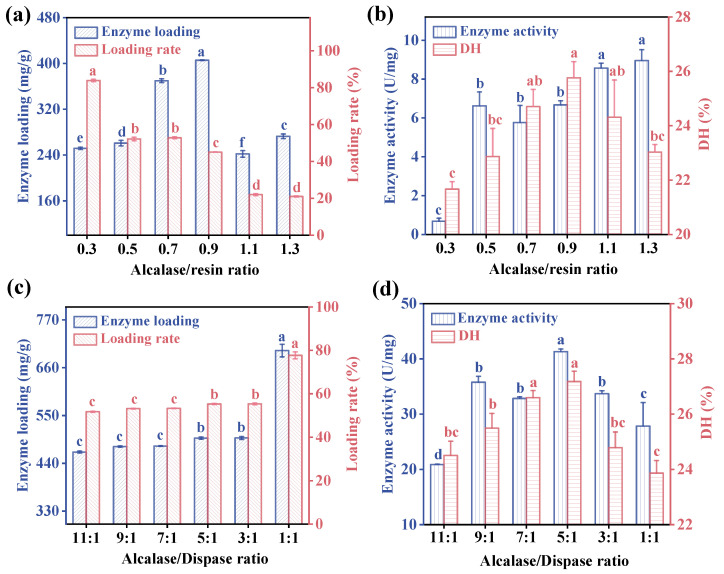
(**a**) The enzyme loading and (**b**) enzyme activity and DH of immobilized Alcalase with different Alcalase/resin ratio. (**c**) The enzyme loading and (**d**) enzyme activity and DH of co-immobilized enzyme under different proportions of two enzymes. Immobilization conditions: temperature = 20 °C, time = 4 h, pH = 9.5 (0.1 M phosphate buffer containing 0.1 M NaCl), and speed = 120 rpm. Different letters indicate significant differences (*p* < 0.05) in the same column.

**Figure 3 foods-13-01753-f003:**
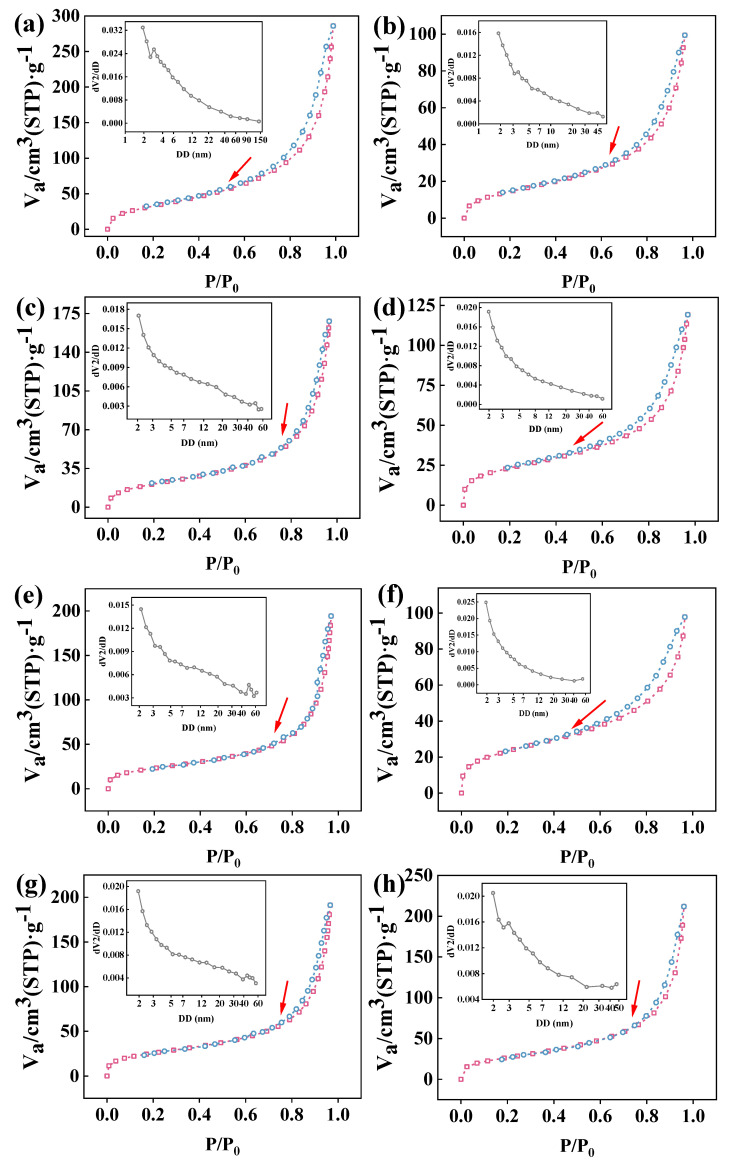
N_2_ absorption–desorption and pore size distribution curve (inset) of samples: (**a**) resin; (**b**) activated resin; (**c**) co-immobilized enzyme_11:1_; (**d**) co-immobilized enzyme_9:1_; (**e**) co-immobilized enzyme_7:1_; (**f**) co-immobilized enzyme_5:1_; (**g**) co-immobilized enzyme_3:1_; (**h**) co-immobilized enzyme_1:1_. Sample pretreatment parameters: T_1_ = 30 °C, t_1_ = 6 min; T_2_ = 90 °C, t_2_ = 30 min; T_3_ = 90 °C, t_3_ = 11 min; T_4_ = 200 °C, t_4_ = 140 min. Red dots: adsorption. Blue dots: desorption.

**Figure 4 foods-13-01753-f004:**
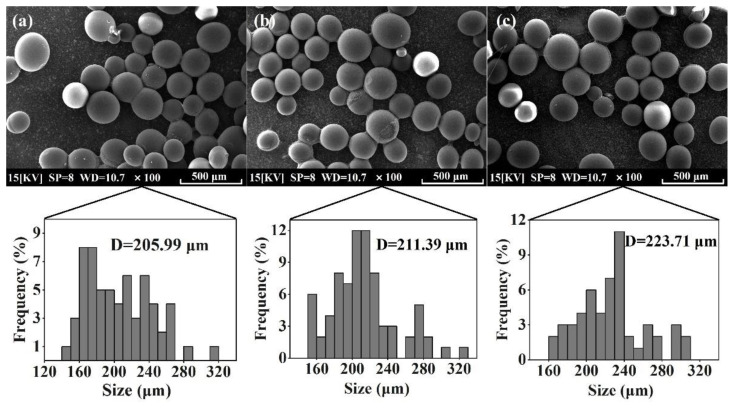
SEM of prepared (**a**) activated resin, (**b**) immobilized Alcalase, and (**c**) co-immobilized enzyme_5:1_.

**Figure 5 foods-13-01753-f005:**
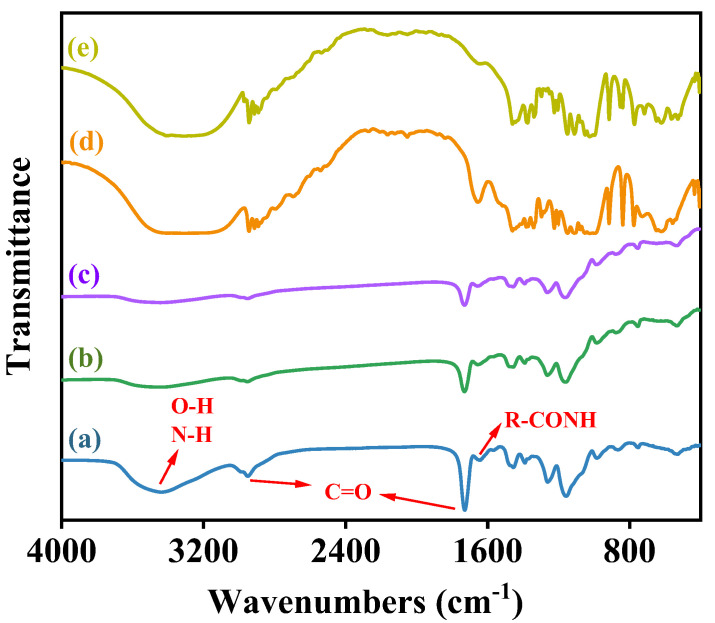
FTIR spectra of (a) activated resin, (b) immobilized Alcalase, (c) Co-immobilized enzyme_5:1_, (d) Alcalase and (e) Dispase.

**Figure 6 foods-13-01753-f006:**
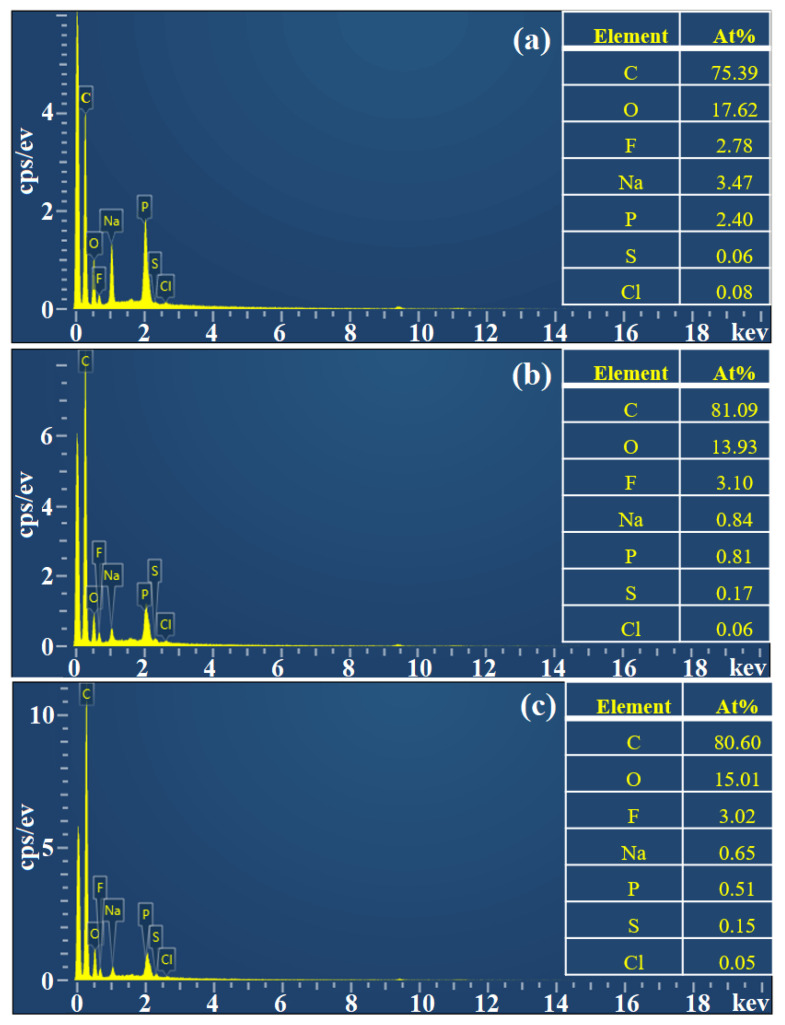
EDX of prepared (**a**) activated resin, (**b**) immobilized Alcalase, and (**c**) co-immobilized enzyme_5:1_.

**Figure 7 foods-13-01753-f007:**
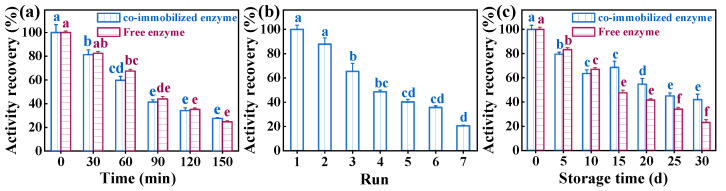
(**a**) The performance of co-immobilized enzyme_5:1_ after storage at 50 °C for a certain period of time. (**b**) Reusability of co-immobilized enzyme_5:1_ for 7 cycles. (**c**) Storage stability of co-immobilized enzyme_5:1_ and free enzyme under the same conditions. Reaction conditions: (**a**) temperature = 50 °C, pH = 9.5 (0.1 M phosphate buffer); (**b**) temperature = 37 °C, pH = 9.5 (0.1 M phosphate buffer); (**c**) temperature = 4 °C, pH = 9.5 (0.1 M phosphate buffer). Different letters indicate significant differences (*p* < 0.05) in the same column.

**Figure 8 foods-13-01753-f008:**
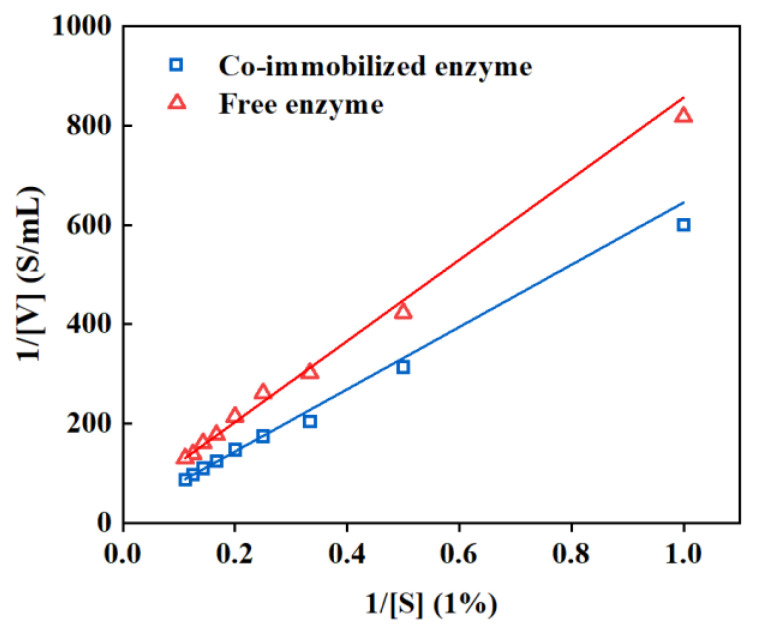
Linear fit based on Lineweaver–Burk. Free enzyme: y = 816.77579 x + 39.6671 (R^2^ = 0.9858, V_m_ = 0.025, K_m_ = 20.59); co-immobilized enzyme_5:1_: y = 626.8223 x + 18.3782 (R^2^ = 0.9879, V_m_ = 0.054, K_m_ = 34.14).

**Figure 9 foods-13-01753-f009:**
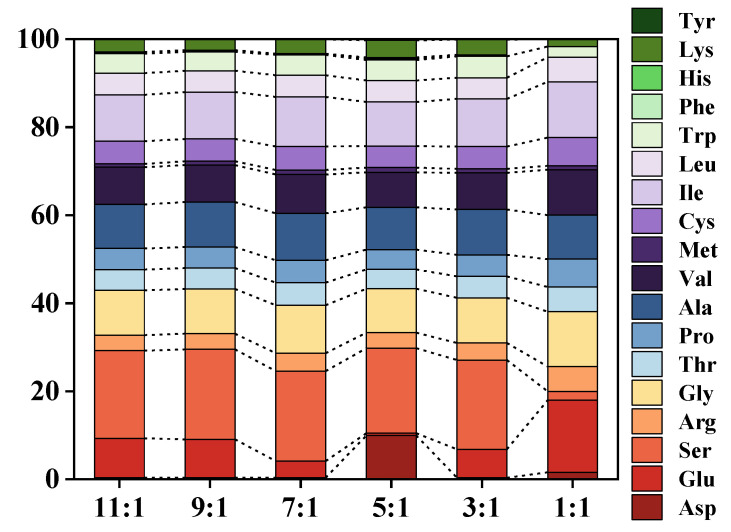
Proportions of 18 kinds of amino acids at different Alcalase/Dispase ratios of co-immobilized enzyme.

**Figure 10 foods-13-01753-f010:**
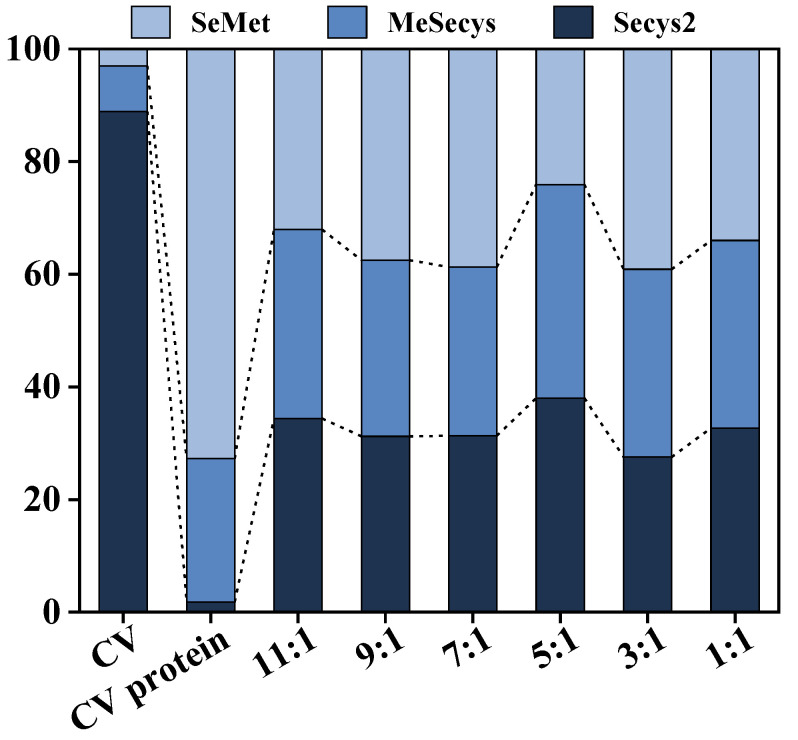
Proportions of Se forms of CV, CV protein, and Se-enriched peptide products at different Alcalase/Dispase ratios of co-immobilized enzyme.

**Table 1 foods-13-01753-t001:** Specific surface area (S_BET_), pore size (D_BJH_), and pore volume (V_total_) of resin, activated resin, and co-immobilized enzyme with different ratio of two enzymes.

Samples	S_BET_/m^2^·g^−1^	D_BJH_/nm	V_total_/cm^3^·g^−1^
Resin	113.678	13.981	0.442
Activated resin	50.261	10.970	0.154
Co-immobilized enzyme_11:1_	75.923	13.072	0.260
Co-immobilized enzyme_9:1_	81.112	8.701	0.184
Co-immobilized enzyme_7:1_	77.231	14.217	0.301
Co-immobilized enzyme_5:1_	81.201	7.065	0.151
Co-immobilized enzyme_3:1_	89.214	12.757	0.296
Co-immobilized enzyme_1:1_	93.074	13.152	0.328

**Table 2 foods-13-01753-t002:** Contents of 18 kinds of amino acids at different Alcalase/Dispase ratio of co-immobilized enzyme.

Amino Acid	11:1 (mg/L)	9:1 (mg/L)	7:1 (mg/L)	5:1 (mg/L)	3:1 (mg/L)	1:1 (mg/L)
Asp	0.041 ± 0.005	0.040 ± 0.006	0.038 ± 0.007	1.126 ± 0.019	0.037 ± 0.006	0.125 ± 0.004
Glu	1.058 ± 0.007	1.11 ± 0.012	0.454 ± 0.014	0.058 ± 0.010	0.737 ± 0.012	1.288 ± 0.012
Ser	2.360 ± 0.006	2.574 ± 0.014	2.452 ± 0.016	2.182 ± 0.017	2.331 ± 0.013	0.160 ± 0.008
Gly	1.211 ± 0.008	1.277 ± 0.005	1.296 ± 0.017	1.131 ± 0.010	1.177 ± 0.017	0.987 ± 0.015
Thr	0.560 ± 0.014	0.602 ± 0.012	0.609 ± 0.015	0.485 ± 0.019	0.559 ± 0.012	0.445 ± 0.013
Pro	0.566 ± 0.016	0.604 ± 0.015	0.610 ± 0.011	0.507 ± 0.007	0.560 ± 0.015	0.504 ± 0.014
Ala	1.187 ± 0.012	1.286 ± 0.013	1.283 ± 0.012	1.085 ± 0.016	1.182 ± 0.012	0.792 ± 0.018
Val	1.01 ± 0.014	1.059 ± 0.017	1.054 ± 0.013	0.898 ± 0.018	0.959 ± 0.011	0.815 ± 0.013
Met	0.090 ± 0.005	0.108 ± 0.009	0.123 ± 0.007	0.117 ± 0.012	0.103 ± 0.007	0.065 ± 0.008
Ile	1.254 ± 0.015	1.346 ± 0.014	1.346 ± 0.011	1.133 ± 0.015	1.237 ± 0.012	1.01 ± 0.012
Leu	0.584 ± 0.014	0.608 ± 0.010	0.589 ± 0.016	0.552 ± 0.005	0.554 ± 0.015	0.438 ± 0.011
Trp	0.531 ± 0.017	0.537 ± 0.013	0.555 ± 0.007	0.529 ± 0.014	0.559 ± 0.012	0.191 ± 0.015
Phe	0.020 ± 0.007	0.022 ± 0.006	0.012 ± 0.003	0.032 ± 0.002	0.017 ± 0.006	
His	0.024 ± 0.006	0.021 ± 0.006	0.016 ± 0.003	0.034 ± 0.002	0.018 ± 0.006	
Lys	0.348 ± 0.005	0.315 ± 0.012	0.390 ± 0.013	0.436 ± 0.012	0.389 ± 0.014	0.129 ± 0.006
Tyr			0.005 ± 0.002	0.028 ± 0.005	0.013 ± 0.003	
Arg	0.415 ± 0.005	0.452 ± 0.014	0.490 ± 0.013	0.399 ± 0.010	0.437 ± 0.010	0.449 ± 0.011
Cys	0.607 ± 0.015	0.637 ± 0.012	0.638 ± 0.012	0.552 ± 0.017	0.577 ± 0.016	0.509 ± 0.010

Notes: Asp = Aspartic, Glu = Glutamic, Ser = Serine, Gly = Glycine, Thr = Threonine, Pro = Proline, Ala = Alanine, Val = Valine, Met = Methionine, Ile = Isoleucine, Leu = Leucine, Trp = Tryptophan, Phe = Phenylalanine, His = Histidine, Lys = Lysine, Tyr = Tyrosine, Arg = Arginine, Cys = Cystine.

**Table 3 foods-13-01753-t003:** Se contents of CV, CV protein, and Se-enriched peptide products at different Alcalase/Dispase ratios of co-immobilized enzyme.

Sample	Total Se	Inorganic Se	Organic Se
CV (mg/kg)	1980.48 ± 50.11	162.77 ± 5.05	1817.70 ± 50.21
CV protein (mg/kg)	1034.45 ± 0.07	53.25 ± 3.20	981.19 ± 3.22
11:1 (mg/L)	10.53 ± 0.21	0.78 ± 0.003	9.75 ± 0.21
9:1 (mg/L)	11.21 ± 0.07	0.81 ± 0.03	10.39 ± 0.05
7:1 (mg/L)	12.91 ± 0.41	0.97 ± 0.10	11.94 ± 0.49
5:1 (mg/L)	12.80 ± 0.11	0.83 ± 0.04	11.98 ± 0.13
3:1 (mg/L)	10.75 ± 0.11	0.93 ± 0.05	9.82 ± 0.13
1:1 (mg/L)	11.17 ± 0.20	0.71 ± 0.06	10.45 ± 0.18

**Table 4 foods-13-01753-t004:** Identified Se-enriched peptides from the enzymatic products of CV proteins at an Alcalase/Dispase ratio of 5:1 for co-immobilized enzymes.

Retention Time	MS (*m*/*z*)	Sequence	Molecular Weight (Da)
1.8075	597.25	SSSSTSKVFLC	1192.4879
18.571	724.36	ILELLLTTYAFC	1446.6913
24.451	718.37	EITVLCDAKVALI	1434.7237
38.923	753.87	CVIASTI	753.3176
40.994	755.38	LVNRKITALCNES	1507.7261
43.816	528.95	LTLEDPTATLEAFLCDKDA	2112.937
44.59	696.87	VLVNRKITALCN	1390.7199
45.586	754.87	VETQLQLFIGLPC	1507.7189
47.201	604.31	SLALCLLSLGGL	1206.6127
48.606	905.92	PKCVSDPL	905.3761
54.373	548.27	LGFVLVCIAL	1094.5643
56.333	623.32	PILVHCKTSAK	1243.6192

**Table 5 foods-13-01753-t005:** Efficiency and application of different immobilized enzymes.

Carrier	Enzyme	Enzyme Loading (mg/g)	Enzyme Active (U/mg)	Loading Rate (%)	Reaction Time (min)	DH (%)	Reference
Resin	Alcalase	405.6	6.7	45.1	120	25.8	This study
Resin	Alcalase and Dispase	497.8	41	77.7	120	27.2	This study
Magnetic beads	Alcalase	925	20.55	45	180	20	[59]
Chitosan	Alcalase	340.2	23.6	96.4	180	29.9	[60]
Alginate beads	Alcalase	671.6	2.7	98.7	225	˂25	[61]
SiO_2_	Alcalase and Flavorzyme	25	180	˂ 80	1080	5.9	[62]

## Data Availability

The data presented in this study are available on request from the corresponding author. The data are not publicly available due to privacy restrictions.

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
