# Peer review of "Co-Immobilization of Alcalase/Dispase for Production of Selenium-Enriched Peptide from Cardamine violifolia"

_foods, 2024, doi:10.3390/foods13111753_

Round 1

Reviewer 1 Report (Previous Reviewer 3)

Comments and Suggestions for Authors

The manuscript is OK and somehow interesting.

However, improvement in the revision is not that significant. I think questions like

Why esepcially this plant ? An Brasscica in general ?

Food safety ? (aspects given are too general) Please explain more. Is this plant safe ?

What about legislation ?

Why is this food ? The way it is written, is more like a technological or bioeconomical study. The main focus is described to be on the immobilized enzymes. Consequently, journal selection still wonders.

Peptides can exert several biological activities ? Are thise helpful ? (besides selen supply) Any additional risks ?

It still needs a clear scientfic hypothesis. (Statements given in the revison such as "By exploring novel preparation techniques for selenium supplements, new avenues are provided for the production of health foods that meet people's needs"are too general.

Is this science or advertising ? It needs more clear statements, more scientific justification and giving limitations (with regard to necessity and safety).

Comments on the Quality of English Language

This is OK.

Author Response

Dear Editor and Referees:

       Thank you for your letter and for the referees’ comments concerning our manuscript entitled “Co-immobilization of Alcalase/Dispase for Production of Selenium-Enriched Peptide from Cardamine violifolia” (manuscript ID: foods-3005816). Main corrections in the revised manuscript and the point-to-point responses to the comments and suggestions of referees are presented as follows. The responses to the comments and suggestions are highlighted in bold, and the corresponding revised parts are marked in red font following the response, as well as in the revised manuscript.

  1. Why esepcially this plant? An Brasscica in general?
  2. Food safety? (aspects given are too general) Please explain more. Is this plant safe?
  3. What about legislation?

Response: We greatly appreciate your comment. Cruciferous plants are widely distributed globally, with particular attention drawn to the Cardamine violifolia (CV) grown in Enshi for its remarkable selenium capabilities. This makes it a suitable candidate for producing selenium peptides with potential health benefits, aiding in addressing selenium deficiency among populations worldwide. Furthermore, this plant is safe. It has long been consumed locally as "wild mustard greens." Moreover, in 2021, it was officially approved by the National Health Commission of the People's Republic of China as a new food ingredient. We have supplemented the description as follows:

Line 57-63:

Cardamine violifolia (CV) is a cruciferous plant with a widespread global distribution. Of notable mention is the CV grown in Enshi, Hubei Province, which is renowned for its remarkable Se enrichment capabilities [9]. In 2021, CV was approved by the National Health Commission of the People's Republic of China as a new food ingredient. As both a medicinal and edible plant, CV possesses rich nutritional value and research significance. Its potential in combating selenium deficiency among populations worldwide is considerable [10].

  1. Why is this food? The way it is written, is more like a technological or bioeconomical study. The main focus is described to be on the immobilized enzymes. Consequently, journal selection still wonders.

Response: We greatly appreciate your comment. Our research is indeed focused on the development and application of enzyme preparations in the field of food science, particularly immobilized enzymes. Immobilized enzymes play a crucial role in enhancing both the efficiency and safety of food processing, ultimately facilitating the development of novel food products with improved nutritional and functional properties. Therefore, we believe our study aligns well with the theme of the "Advances in Food Enzymology: Development and Application of Enzyme Preparations" Special Issue of Foods.

  1. Peptides can exert several biological activities ? Are thise helpful ? (besides selen supply) Any additional risks?
  2. Is this science or advertising? It needs more clear statements, more scientific justification and giving limitations (with regard to necessity and safety).

Response: Thanks for your helpful suggestion. In addition to Se activity, peptides can provide a variety of biological activities including antioxidant, anti-inflammatory, antibacterial and even anticancer properties depending on the sequence and structure of the peptide. The safety and efficacy of Se-enriched peptides from CV have been verified in many literatures. Our study introduced new co-immobilized enzymes to scientifically investigate the feasibility of enzymatic preparation of Se-enriched peptides from CV. We have added the necessary notes and literature to emphasize the necessity and safety of this study as follows:

Line 65-71:

  1. Du et al. [11] investigated the bioavailability of total Se and five Se compounds in CV leaves using an in vitro digestion combined with a Caco-2 cell monolayer model and found that the bioavailability of total Se was 49.11%, and that the bioavailability of MeSeCys and SeMet was higher than that of SeCys2. It has been found that Se in CV is predominantly found in Se-enriched proteins, which have a higher content of SeCys compared to SeMet [12], thus it is crucial to explore the Se forms of the different active constituents in CV.

Line 75-79:

  1. Ling et al. [14] prepared Se-enriched peptides by enzymatic hydrolysis of CV proteins using Alcalase and Dispase, and found that Se-enriched peptides increased mRNA expression of GPX1 and NGFR in ethanol-treated L-02 hepatocytes, confirming their protective effect against ethanol-induced liver injury.

Line 548-549:

  1. Du, C.; Wang, P.; Li, Y.; Cong, X.; Huang, D.; Chen, S.; Zhu, S. Investigation of selenium and selenium species in Cardamine violifolia using in vitro digestion coupled with a Caco-2 cell monolayer model. Food Chem. 2024, 444, 138675-138684.

Line 555-556:

  1. Ling, R.; Du, C.; Li, Y.; Wang, S.; Cong, X.; Huang, D.; Chen, S.; Zhu, S. Protective Effect of Selenium-enriched Peptide from Cardamine violifolia on Ethanol-induced L-02 Hepatocyte Injury. Biol. Trace Elem. Res. 2024.

  1. It still needs a clear scientfic hypothesis. (Statements given in the revison such as "By exploring novel preparation techniques for selenium supplements, new avenues are provided for the production of health foods that meet people's needs"are too general.

Response: Thanks for your helpful suggestion. We have revised the original text and supplemented the description as follows:

Line 104-108:

Herein, the research aimed to investigate the feasibility of co-immobilized enzyme hydrolysis for producing Se-enriched peptides from CV protein. The hydrolysis process of CV protein was studied using the co-immobilization technique of two enzymes, resulting in Se-enriched peptides with uniform distribution of Se forms and amino acid types. This provides additional possibilities for the selection of Se supplements.

Reviewer 2 Report (Previous Reviewer 2)

Comments and Suggestions for Authors

Thank the authors for their answers and modifications. These improved the manuscript. I have remained only three remarks.

1. Tertiary amines do not react with aldehydes. How were the enzymes covalently linked to the resin?

2. Based on the Figure 2 the Enzyme loading values did not change in the revised manuscript. Were these values recalculated using the new formula 2?

3. During the enzymatic hydrolysis mainly peptides are formed.  Thus the answer for my question 9 does not explain how the different Se forms (SeMet, MeSecys, Secys2) were produced.

Author Response

  1. Tertiary amines do not react with aldehydes. How were the enzymes covalently linked to the resin?

Response: We greatly appreciate your comment. The carrier we selected is a resin with amino groups on the surface. The carrier surface was activated using glutaraldehyde functional reagent, and the amino groups on the carrier could be covalently linked to the functional groups of amino acid residues on the enzyme, achieving the immobilization of the enzyme onto the resin.

  1. Based on the Figure 2 the Enzyme loading values did not change in the revised manuscript. Were these values recalculated using the new formula 2?

Response: Thanks for your comments. Actually, our previous version was calculated using the new formula, but there was an error in the formula when we initially drafted the manuscript. Therefore, we just corrected the formula, and the values of enzyme loading remained unchanged.

  1. During the enzymatic hydrolysis mainly peptides are formed. Thus the answer for my question 9 does not explain how the different Se forms (SeMet, MeSecys, Secys2) were produced.

Response: Thanks for your comments. In the structure of proteins, sulfur atoms are typically found in sulfur-containing amino acids such as cysteine and methionine. Selenium, being a homologous element to sulfur, can replace sulfur in amino acids during the enzymatic hydrolysis of Cardamine violifolia proteins, forming seleno-amino acids such as selenocysteine and selenomethionine. Additionally, since different enzymes have distinct cleavage sites, we can explore the influence of various co-immobilized enzymes on the selenium forms.

Reviewer 3 Report (New Reviewer)

Comments and Suggestions for Authors

The study presented points out some differences from related research. It underlines the possibilities of using co-immobilized selenium-enriched peptides from Cardamine violifolia. The authors included comparative information between their results and those from reference literature. The findings could be valuable in developing novel nutraceutical products. The information presented may be relevant in different domains, such as pharmacy and/or food.

Since the comprehensive data presented were obtained in vitro conditions, could the authors mention if similar research has already been done in vivo environments or simulated ones (e.g., artificial digestion conditions)? If the literature does not provide this information, could they mention some limitations of such a scientific approach based on valuable literature data?

It has to be revised by the authors and resubmitted with suggested modifications specified in the reviewer’s comments.

Comments on the Quality of English Language

Minor editing of English language required.

Author Response

Dear Editor and Referees:

       Thank you for your letter and for the referees’ comments concerning our manuscript entitled “Co-immobilization of Alcalase/Dispase for Production of Selenium-Enriched Peptide from Cardamine violifolia” (manuscript ID: foods-3005816). Main corrections in the revised manuscript and the point-to-point responses to the comments and suggestions of referees are presented as follows. The responses to the comments and suggestions are highlighted in bold, and the corresponding revised parts are marked in blue font following the response, as well as in the revised manuscript.

The study presented points out some differences from related research. It underlines the possibilities of using co-immobilized selenium-enriched peptides from Cardamine violifolia. The authors included comparative information between their results and those from reference literature. The findings could be valuable in developing novel nutraceutical products. The information presented may be relevant in different domains, such as pharmacy and/or food.

Since the comprehensive data presented were obtained in vitro conditions, could the authors mention if similar research has already been done in vivo environments or simulated ones (e.g., artificial digestion conditions)? If the literature does not provide this information, could they mention some limitations of such a scientific approach based on valuable literature data? It has to be revised by the authors and resubmitted with suggested modifications specified in the reviewer’s comments.

Response: Thanks for your helpful suggestion. We have added the relevant research notes in the main text as follows:

Line 65-71:

  1. Du et al. [11] investigated the bioavailability of total Se and five Se compounds in CV leaves using an in vitro digestion combined with a Caco-2 cell monolayer model and found that the bioavailability of total Se was 49.11%, and that the bioavailability of MeSeCys and SeMet was higher than that of SeCys2.

Line 75-79:

  1. Ling et al. [14] prepared Se-enriched peptides by enzymatic hydrolysis of CV proteins using Alcalase and Dispase, and found that Se-enriched peptides increased mRNA expression of GPX1 and NGFR in ethanol-treated L-02 hepatocytes, confirming their protective effect against ethanol-induced liver injury.

Line 548-549:

  1. Du, C.; Wang, P.; Li, Y.; Cong, X.; Huang, D.; Chen, S.; Zhu, S. Investigation of selenium and selenium species in Cardamine violifolia using in vitro digestion coupled with a Caco-2 cell monolayer model. Food Chem. 2024, 444, 138675-138684.

Line 555-556:

  1. Ling, R.; Du, C.; Li, Y.; Wang, S.; Cong, X.; Huang, D.; Chen, S.; Zhu, S. Protective Effect of Selenium-enriched Peptide from Cardamine violifolia on Ethanol-induced L-02 Hepatocyte Injury. Biol. Trace Elem. Res. 2024.

Reviewer 4 Report (New Reviewer)

Comments and Suggestions for Authors

The manuscript foods-3005816 investigated the co-immobilization of alcalase/dispase for production of selenium-enriched peptide from Cardamine violifolia.

Materials section:

Include the Enzyme Commission number (EC number) of alcalase/dispase. EC is important as it is a numerical classification scheme for enzymes based on the chemical reactions they catalyze.

Unit “katal” must be used for catalytic activity according to IUPAC (https://publications.iupac.org/pac/pdf/2001/pdf/7306x0927.pdf)

The sentence ‘CV was acquired from Enshi [22]’ requires clarification. The source of the material needs to be explicitly stated.

Results section:

Figure 9 must be improved. In the present form, the interpretation of figures is complicated.

In Tables 1,2,3 and 5, the standard deviations must be included.

Conclusions section:

It is essential to recommend new areas for future research.

Author Response

Dear Editor and Referees:

       Thank you for your letter and for the referees’ comments concerning our manuscript entitled “Co-immobilization of Alcalase/Dispase for Production of Selenium-Enriched Peptide from Cardamine violifolia” (manuscript ID: foods-3005816). Main corrections in the revised manuscript and the point-to-point responses to the comments and suggestions of referees are presented as follows. The responses to the comments and suggestions are highlighted in bold, and the corresponding revised parts are marked in purple font following the response, as well as in the revised manuscript.

  1. Materials section:

Include the Enzyme Commission number (EC number) of alcalase/dispase. EC is important as it is a numerical classification scheme for enzymes based on the chemical reactions they catalyze.

Response: Thanks for your valuable suggestion. I have added the EC numbers for Alcalase and Dispase in the original text as follows:

Line: 121-123:

Alcalase (2 × 106 U/g, BRENDA: EC3.4.21.62) from Bacillus licheniformis and Dispase (1 × 106 U/g, BRENDA: EC3.4.24.28) from Bacillus subtilis were supplied by Solarbio Co., Ltd. (Beijing, China).

  1. Unit “katal” must be used for catalytic activity according to IUPAC (https://publications.iupac.org/pac/pdf/2001/pdf/7306x0927.pdf)

Response: Thank you for your comment. We chose to use U/mg as the unit because our study primarily focuses on the activity of co-immobilized enzymes on specific matrices (resins). In contrast, the use of the katal unit would overlook the influence of the matrix mass. Additionally, in the Results and Discussion section (Table 5), we sought relevant literature for comparison and found that the units for catalytic activity of immobilized enzymes, particularly in the preparation of peptides by enzyme hydrolysis, were mainly expressed as U/mg (units of activity per milligram of matrix). Therefore, we believe this unit is more suitable for describing the activity level of immobilized enzymes on given matrices, facilitating a direct comparison of different immobilized enzyme samples.

  1. The sentence ‘CV was acquired from Enshi [22]’ requires clarification. The source of the material needs to be explicitly stated.

Response: Thank you for your comment. The material was acquired from Enshi Se-Run Material Engineering Technology Co., Ltd. (Hubei, China), and we have explicitly stated this in the manuscript.

Line 133-134:

CV was acquired from Enshi Se-Run Material Engineering Technology Co., Ltd. (Hubei, China) [23].

  1. Results section:

Figure 9 must be improved. In the present form, the interpretation of figures is complicated.

Response: Thanks for your helpful suggestion. We have improved Figure 9 to enhance its clarity and interpretability. Additionally, to maintain consistency, we have also made corresponding changes to Figure 10 as follows:

Line 460-461:

Figure 9. Proportions of 18 kinds of amino acids at different Alcalase/Dispase ratios of co-immobilized enzyme.

Line 482-484:

Figure 10. Proportions of Se forms of CV, CV protein, and Se-enriched peptide products at different Alcalase/Dispase ratios of co-immobilized enzyme.

  1. In Tables 1,2,3 and 5, the standard deviations must be included.

Response: We greatly appreciate your comment.

1) We are sorry that due to the nature of the test and calculation methods themselves, we cannot provide the standard deviations for Table 1. Firstly, we measured the nitrogen adsorption and desorption data of the samples at different relative pressures and plotted the graphs of adsorption and desorption isotherms. Subsequently, the isotherms were fitted using the BET equation, and data such as the specific surface area and pore size of the samples were determined by fitting multiple points in the experimental data. In fact, several repeated experiments were performed to verify the reliability of the results, but to ensure the consistency and rigor of the data in Table 1 and Fig. 3, we believe that the standard deviations should not be given here.

2) Regarding the standard deviations of Tables 2 and 3, we are sorry that they were not given before to maintain the consistency of the tables. Thank you again for your suggestion and we have added the relevant data.

3) Table 5 compares the efficiency of immobilized enzymes in our and other literature. Some of the literature does not give the standard deviation, and some only give the graphs, so we are sorry that we cannot add this part of the data.

Line 463-464

Table 2. Contents of 18 kinds of amino acids at different Alcalase/Dispase ratio of co-immobilized enzyme.

Amino acid

11:1 (mg/L)

9:1 (mg/L)

7:1 (mg/L)

5:1 (mg/L)

3:1 (mg/L)

1:1 (mg/L)

Asp

0.041±0.005

0.040±0.006

0.038±0.007

1.126±0.019

0.037±0.006

0.125±0.004

Glu

1.058±0.007

1.11±0.012

0.454±0.014

0.058±0.010

0.737±0.012

1.288±0.012

Ser

2.360±0.006

2.574±0.014

2.452±0.016

2.182±0.017

2.331±.0.013

0.160±0.008

Gly

1.211±0.008

1.277±0.005

1.296±0.017

1.131±0.010

1.177±0.017

0.987±0.015

Thr

0.560±0.014

0.602±0.012

0.609±0.015

0.485±0.019

0.559±0.012

0.445±0.013

Pro

0.566±0.016

0.604±0.015

0.610±0.011

0.507±0.007

0.560±0.015

0.504±0.014

Ala

1.187±0.012

1.286±0.013

1.283±0.012

1.085±0.016

1.182±0.012

0.792±0.018

Val

1.01±0.014

1.059±0.017

1.054±0.013

0.898±0.018

0.959±0.011

0.815±0.013

Met

0.090±0.005

0.108±0.009

0.123±0.007

0.117±0.012

0.103±0.007

0.065±0.008

Ile

1.254±0.015

1.346±0.014

1.346±0.011

1.133±0.015

1.237±0.012

1.01±0.012

Leu

0.584±0.014

0.608±0.010

0.589±0.016

0.552±0.005

0.554±0.015

0.438±0.011

Trp

0.531±0.017

0.537±0.013

0.555±0.007

0.529±0.014

0.559±0.012

0.191±.015

Phe

0.020±0.007

0.022±0.006

0.012±0.003

0.032±0.002

0.017±0.006

His

0.024±0.006

0.021±0.006

0.016±0.003

0.034±0.002

0.018±0.006

Lys

0.348±0.005

0.315±0.012

0.390±0.013

0.436±0.012

0.389±0.014

0.129±0.006

Tyr

0.005±0.002

0.028±0.005

0.013±0.003

Arg

0.415±0.005

0.452±0.014

0.490±0.013

0.399±0.010

0.437±0.010

0.449±0.011

Cys

0.607±0.015

0.637±0.012

0.638±0.012

0.552±0.017

0.577±0.016

0.509±0.010

Line 480-481

Table 3. Se Contents of CV, CV protein, and Se-enriched peptide products at different Alcalase/Dispase ratios of co-immobilized enzyme.

Sample

Total Se

Inorganic Se

Organic Se

CV (mg/kg)

1980.48±50.11

162.77±5.05

1817.70±50.21

CV protein (mg/kg)

1034.45±0.07

53.25±3.20

981.19±3.22

11:1 (mg/L)

10.53±0.21

0.78±0.003

9.75±0.21

9:1 (mg/L)

11.21±0.07

0.81±0.03

10.39±0.05

7:1 (mg/L)

12.91±0.41

0.97±0.10

11.94±0.49

5:1 (mg/L)

12.80±0.11

0.83±0.04

11.98±0.13

3:1 (mg/L)

10.75±0.11

0.93±0.05

9.82±0.13

1:1 (mg/L)

11.17±0.20

0.71±0.06

10.45±0.18

  1. Conclusions section:

It is essential to recommend new areas for future research.

Response: Thank you for your valuable suggestion. We have made the necessary additions in the corresponding section.

Line 512-515:

This study provides novel insights into the field of enzymatic immobilization for plant protein hydrolysis. Future research will involve the utilization of this co-immobilized enzyme to prepare Se-enriched antioxidant peptides, and investigate their biological effects in cells.

Round 2

Reviewer 1 Report (Previous Reviewer 3)

Comments and Suggestions for Authors

However, I am not sure if this is renitency, misunderstanding, or just a question of different mentalities, but authors did not answer/discuss a single aepct of the last review round.They just give nice statement, but without improving. This is not the scientific way...

I will try again:

Why was especcially this plant used for selenium enrichment ? It is originally healthy. So why needs it to be improved ? Why not taking an other plant ? Please explain the text.

In this context: Does it necessarily must be a Brasscica for enriching ? Would another plant family also work ? Please explain the text.

When changing the properties of a food (plant), food safety need to be clear. You will not take the risk to make on the hand more healthier, but on the other hand some aspects that you cannot take into account (e.g., metabolism of the plant after enriching) that will make the plant less healthier will occur too. Is this plant safe after the modification ? Please discuss in the text.

In most of the world's countries, there are legal requirements when a food plant is modified. Does this aspect apply here too ? What about legislation ? Please discuss in the text.

Is this really food science ? It appears that it does not look like food science. Consequently, it needs to be revised that it will look like food science. This would be necessary for the Special Issue, as the journal will be stay the same. The way it is written now, is more like a technological or bioeconomical study. The main focus is described to be on the immobilized enzymes. Consequently, journal selection still wonders. In the end I feel confused why the selen topic and the immobilized enzyme topic have been mixed. Why was this not separated or .......... far better explained at hand of clear scientfic hypotheses ?

As mentioned before, peptides can exert some many biological activities ?can there be any side effects ? Any further risks ? For example, when such a plant is grown, will it be a risk for the other sourrounding plants ?

It still needs a clear scientfic hypothesis. Statements given are too general.

Science is about having an idea, form a hypothesis, form specific aims and follow afterwards these aims. Even more important to consider the outcomes for risks and limitations.

Working in natural sciences is about accuracy, carefulness, and a steady state of reconsidering what one is doing/writing. Ideally, taking a all the world's people and the whole planet into account.

Is this here science or advertising ? It needs more clear statements, more scientific justification and giving limitations (with regard to necessity and safety). Who will take advantage of ths research, but who will also maybe suffer from this ?

Figure 3 is by far too small. Who should read that ?

Labelling of Figure 4 is poor. Cannot be read properly.

The same for Figure 6.

The same fo Figure 7. Labelling of the x-axes are neither on the same height nor do they have the same font size.

It is all a question of accuracy and carefulness....

Overall: This still very confusing....both topics are quite good, but the way of writing seems to be not very traditional and leads to confusion.

Main scientific aspects are disrespected: Hypotheses, Justifying necessities, Discussing limitations, Discussing risks, Making accurate figures by initially thinking from a readers's perspective, Making accurate figures in general (just for the pure beauty).

Comments on the Quality of English Language

This is more or less OK.

Author Response

Dear Editor and Referees:

       Thank you for your letter and for the referees’ comments concerning our manuscript entitled “Co-immobilization of Alcalase/Dispase for Production of Selenium-Enriched Peptide from Cardamine violifolia” (manuscript ID: foods-3005816). Main corrections in the revised manuscript and the point-to-point responses to the comments and suggestions of referees are presented as follows. The responses to the comments and suggestions are highlighted in bold, and the corresponding revised parts are marked in red font following the response, as well as in the revised manuscript.

1. Why was especcially this plant used for selenium enrichment ? It is originally healthy. So why needs it to be improved ? Why not taking an other plant ? Please explain the text.

Response: Thank you for your valuable suggestion. We must state a fact: our study did not involve additional selenium enrichment in plants. In 1994, the Geological Department of Hubei Province discovered the "only independently proven Se deposit in the world" in Enshi City, followed by the discovery of the super Se-enriched plant Cardamine violifolia (with a Se content of nearly 2000 mg/kg) growing there. In 2021, the National Health Commission of the People's Republic of China approved Cardamine violifolia as a new food raw material. Our study only used this Se-enriched plant, Cardamine violifolia, as a raw material and employed co-immobilized enzymes to hydrolyze Se-enriched protein from Cardamine violifolia to prepare Se-enriched peptides. In order to better elucidate the safety of Cardamine violifolia, we have listed the current Chinese standards for its use:

1) T/HBSE 0017-2023 (Cardamine violifolia peptide powder)

2) T/HBSE 0016-2023 (Cardamine violifolia protein powder)

3) Q/ESXS 0018S-2022 (Cardamine violifolia powder)

4) Q/LPX 0002S-2022 (Se-enriched Cardamine violifolia powder)

2. In this context: Does it necessarily must be a Brasscica for enriching? Would another plant family also work? Please explain the text.

Response: Thank you for your valuable suggestion. We chose Cardamine violifolia (a member of the Brassicaceae family) for our study based on its well-documented ability to accumulate high levels of Se from Se-enriched soils and its nutritional properties. The co-immobilized enzymes developed in this study can be used for the hydrolysis of plant proteins, not limited to Brassicaceae plants. Additionally, Se enrichment does not strictly require Brassicaceae plants; however, we selected this plant due to its natural Se-enriched characteristics. This choice allows us to better utilize its inherent high Se levels to explore the effects of different enzyme ratios on the production of Se-enriched peptides through co-immobilized enzyme hydrolysis.

3. When changing the properties of a food (plant), food safety need to be clear. You will not take the risk to make on the hand more healthier, but on the other hand some aspects that you cannot take into account (e.g., metabolism of the plant after enriching) that will make the plant less healthier will occur too. Is this plant safe after the modification ? Please discuss in the text.

In most of the world's countries, there are legal requirements when a food plant is modified. Does this aspect apply here too ? What about legislation ? Please discuss in the text.

Response: Thank you for your valuable suggestion. Cardamine violifolia has been developed as a source of medicinal and edible products, which we can consume as Se supplements [1]. We chose Cardamine violifolia also because of its natural Se-enriched properties, and we did not make any genetic or chemical modifications to the plant itself. [2-3]. Therefore, there are no food safety concerns related to such modifications. Our research primarily focuses on the hydrolysis of Se-enriched plant proteins using co-immobilized enzymes to produce Se-enriched peptides, which does not alter the genetic or chemical composition of the plant.

  1. Guo Z, Zhu B, Guo J, et al. Impact of selenium on rhizosphere microbiome of a hyperaccumulation plant Cardamine violifolia[J]. Environmental Science and Pollution Research, 2022, 29(26): 40241-40251.
  2. Li J, Huang C, Lai L, et al. Selenium hyperaccumulator plant Cardamine enshiensis: from discovery to application[J]. Environmental Geochemistry and Health, 2023, 45(8): 5515-5529.
  3. Xu X, Wei Y, Zhang Y, et al. A new selenium source from Se-enriched Cardamine violifolia improves growth performance, anti-oxidative capacity and meat quality in broilers[J]. Frontiers in Nutrition, 2022, 9: 996932.

4. Is this really food science ? It appears that it does not look like food science. Consequently, it needs to be revised that it will look like food science. This would be necessary for the Special Issue, as the journal will be stay the same. The way it is written now, is more like a technological or bioeconomical study. The main focus is described to be on the immobilized enzymes. Consequently, journal selection still wonders. In the end I feel confused why the selen topic and the immobilized enzyme topic have been mixed. Why was this not separated or .......... far better explained at hand of clear scientfic hypotheses ?

Response: Thank you for your valuable suggestion. Our research indeed falls within the field of food science, and we did not carry out additional Se enrichment. Instead, we developed a method to produce Se-enriched peptides from Cardamine violifolia proteins using co-immobilized enzymes. Although one of the main focuses of our study is the development and application of co-immobilized enzymes, this is closely related to our goal of preparing Se-enriched peptides with uniform Se forms distribution and complete amino acid composition through this method. To better align with the theme of food science, we have clarified the following points in the manuscript:

1) Application Background in Food Science: We have added relevant literature in the manuscript, elaborating on the health benefits of Se-enriched peptides as functional food ingredients and their potential applications in the food industry.

2) Scientific Hypotheses: We have supplemented the scientific hypotheses of the study, namely that immobilized enzyme technology can effectively improve the production efficiency and quality of peptides.

Line 79-82:

1. Zhao et al. [15] prepared egg yolk selenium-enriched peptides using Alcalase and Dispase hydrolysis combined with shear pretreatment, which exhibited excellent immunoassays, including immune organ index, serum levels of immune factors, splenic histopathological changes, and hepatic selenium content.

Line 573-575:

15. Zhao, Q.; McClements, D. J.; Li, J.; Chang, C.; Su, Y.; Gu, L.; Yang, Y. Egg Yolk Selenopeptides: Preparation, Characterization, and Immunomodulatory Activity. J. Agric. Food Chem. 2024, 72(10), 5237–5246.

Line 105-108:

Furthermore, Y. Wang et al. [24] showed that immobilized dual enzyme exhibited a higher zein DH (65.8%) compared to free enzyme (49.3%) and single enzyme immobilized in calcium alginate beads (45.5%).

Line 600-601:

24. Wang, Y.; Chen, H.; Wang, J.; Xing, L. Preparation of active corn peptides from zein through double enzymes immobilized with calcium alginate–chitosan beads. Process Biochem. 2014, 49(10), 1682-1690.

5. As mentioned before, peptides can exert some many biological activities ?can there be any side effects ? Any further risks ? For example, when such a plant is grown, will it be a risk for the other sourrounding plants ?

Response: Thank you for your concern. We have cited numerous references before in the manuscript demonstrating the safety and efficacy of Se-enriched peptides from Cardamine violifolia, including their protective effects against ethanol-induced liver cell damage and high-fat diet-induced obesity and related metabolic disorders in mice [1-2]. Furthermore, the Cardamine violifolia used in our study is naturally grown without any artificial Se intervention, thus posing no risk to other plants.

  1. Ling R, Du C, Li Y, et al. Protective Effect of Selenium-enriched Peptide from Cardamine violifolia on Ethanol-induced L-02 Hepatocyte Injury[J]. Biological Trace Element Research, 2024: 1-14.
  2. Protective effects of selenium-enriched peptides from Cardamine violifolia against high-fat diet induced obesity and its associated metabolic disorders in mice.

6. It still needs a clear scientfic hypothesis. Statements given are too general. Science is about having an idea, form a hypothesis, form specific aims and follow afterwards these aims. Even more important to consider the outcomes for risks and limitations. Working in natural sciences is about accuracy, carefulness, and a steady state of reconsidering what one is doing/writing. Ideally, taking a all the world's people and the whole planet into account. Is this here science or advertising ? It needs more clear statements, more scientific justification and giving limitations (with regard to necessity and safety). Who will take advantage of ths research, but who will also maybe suffer from this ?

Response: Thank you for your helpful suggestions. We have rewritten the Introduction section to emphasize the hypothesis of this study. In addition, the limitation of this study is that the activity of the prepared co-immobilized enzyme, although already high, is still not as good as that of the free enzyme, which may be attributed to the fact that during the immobilization process, some of the amino acid residues of the active centers were involved in the binding with the carrier, leading to the loss of activity of some of the enzyme, which has been added in the Results section. Moreover, cellular experiments to test the safety of the Se-enriched peptides prepared from this co-immobilized enzyme have not been addressed in this paper, and this section is also added in the Conclusion section.

Line 111-124:

Herein, the research aimed to investigate the feasibility of co-immobilized enzyme hydrolysis for producing Se-enriched peptides from CV protein. Co-immobilization of two enzymes (Alcalase and Dispase) was used to study the hydrolysis of CV proteins, which ultimately produced Se-enriched peptides with uniform Se forms and amino acid distribution. Alcalase and Dispase were chosen for their distinct recognition sites, which are expected to have a synergistic effect during enzyme immobilization, thus enhancing the overall co-immobilized enzyme activity. The resin was chosen as a carrier because its abundant amino groups could provide suitable immobilization sites for various enzymes, thereby enhancing the efficiency and activity of enzyme immobilization. In this study, the structure and activity of different co-immobilized enzymes were explored by adjusting the ratio of Alcalase and Dispase. On this basis, the effects of such changes on the amino acid composition and Se morphology distribution of the hydrolysis products were analyzed in detail. The results of this research are expected to provide more options and ideas in the field of hydrolysis of plant proteins by co-immobilized enzymes.

Line 504-507:

However, it is important to note that although these co-immobilized proteases showed high hydrolysis under hydrolysis of plant proteins, the limitations of the immobilization process that may lead to a decrease in enzyme activity need to be addressed by further studies.

Line 522-524:

Future research needs to further use co-immobilized enzymes with high enzyme activity to prepare Se-enriched antioxidant peptides, investigate their safety, and study their biological effects in cells.

7. Figure 3 is by far too small. Who should read that ? Labelling of Figure 4 is poor. Cannot be read properly.

The same for Figure 6.

The same fo Figure 7. Labelling of the x-axes are neither on the same height nor do they have the same font size.

It is all a question of accuracy and carefulness....

Overall: This still very confusing....both topics are quite good, but the way of writing seems to be not very traditional and leads to confusion.

Main scientific aspects are disrespected: Hypotheses, Justifying necessities, Discussing limitations, Discussing risks, Making accurate figures by initially thinking from a readers's perspective, Making accurate figures in general (just for the pure beauty).

Response: Thanks for your comment. We are sorry for the previous graphs. We have reworked the graph for better readability. For the hypotheses and limitations of the text, we have made additions in the previous question.

Line 375 (Figure 3):

Line 405 (Figure 4):

Line 412 (Figure 6):

Line 430 (Figure 7):

This manuscript is a resubmission of an earlier submission. The following is a list of the peer review reports and author responses from that submission.

Round 1

Reviewer 1 Report

Comments and Suggestions for Authors

It is valuable and worth of publishing report. My comments are mostrly of technical nature (with one exception).

General comment to paragraph 3.4.:

This part is unclear because results relate to amino acid composition of digestet product. Does it contain only small peptides? What is removed after hydrolysis (protein)?
What is the source of differences in amino acid compositions? Are there also products of hydrolysis of alcalase and dispase?

Specific comments:

1./ references should be ordered in terms of journal names (Foods usually requires abbreviations of the journal names) and names of the organisms in the titles (usually are given in italics by the cited autors);

2./ Names of organisms should be given in italics;

3./ Cardamine violifolia was not discovered in Enshi - it is a popular plant all over the world. It also grows in Enshi (line 45).

4./ names of enzymes should be given uniformy. For example (line 70) -  if Alcalase then Papain; if papain then alcalase;

5./ something is lost in first sentence of paragraph 2.7.1.;

6./ it would be beneficial to compare the porosity of the modified resin with porosity of substrate resin (paragraph 3.2.);

7./ Figure 4: blue on black background is not well visible. It would be beneficial to make the background of insertion white;

8./ t is better to use "immobilized alcalase" instead of "immobilized enzyme" through the text.

Comments on the Quality of English Language

English is o.k. to me

Reviewer 2 Report

Comments and Suggestions for Authors

In this manuscript the authors describe the effect of immobilized enzymes on the digestion of Se containing proteins of Cardamine violifolia (CV). The co-immobilized Alcalase and Dispase enzymes produced Se enriched peptides that can serve as valuable selenium supplements. The effect of the solid-to-liquid ratio (S/L), enzyme-to-substrate ratio (E/S), pH and the ratio of two enzymes on the degree of hydrolysis (DH) was studied.

Remarks and questions:

1. The first sentence of the introduction should be changed.

2. I could not find the name of Dispase enzyme in the reference 18. Which enzyme is corresponding to Dispase. If none of them, better reference should be given. For better understanding short description of the two enzymes, optimal pH range, cleavage sites, endo- or exopeptidase should be given in the introduction part.

3. I could not find the resin at the distributor’ site. Thus it is hard to identify the type and properties of the used resin. It should be described in the manuscript.

4. In chapter 2.2 a 3.5 isoelectric point is given. Whose pI is this? During the dialysis were the precipitate dissolved? The mass of isolated CV protein should be given here.

5. Why alkali-soluble proteins were isolated? Is there any reference that this fraction has the most Se-containing proteins?

6. In chapter 2.3 the missing experimental parameters should be given (buffer if it was used, its concentration, the mass of CV protein, volume of solvent, the concentration of enzyme)

7. The chemical formula of Na2HPO4 is misspelled in many places.

8. In chapter 2.4.1 what was the solvent of GA?

It is not clear how the amount of Tyr was determined. If it was not separated from the peptide fragments how was it distinguished from other aromatic amino acids.

In formula 1 why were the concentrations used instead of the mass of proteins? Was the volume of solutions the same? This parameter should be given.

9. In chapter 2.7.2 it is not clear from what were the Se-containing amino acids separated? Were proteins and peptides hydrolysed first?

10. In case of different S/L was the CV protein always soluble?

11. Why is there any effect of the S/L on the substrate-enzyme binding (line 252)? If the concentration of the enzyme is the same in all S/L it may have a limit in DH which is independent of the amount of proteins after a threshold.

12. In case of all figures the main experimental parameters should be given in the figure caption for the better understanding.

13. How could it be explained that the increases in the amount of enzyme does not increase the DH (line 259)? Maybe if the number of cleavable bonds is limited in the CV protein, above an enzyme concentration the enzyme concentration has not any effect.

14. In Figure 2a it is not clear why the increased amount of enzyme did not increase the enzyme loading at 0.5 ratio, but increased at 0.7 ratio? It is not also clear if the loading reaches a maximum (at 0.9 ratio) what is the effect which inhibits to get this loading at higher ratio?

What is the reason that the enzyme loading does not correlate to the enzyme activity and enzyme activity does not correlate with DH (Figure 2b)?

15. Is the Dispase a nucleic acid exonuclease (line 380)?

16. In table 5 it would be worth giving the Se content of digested products.

16. Based on Figure 10 and Table 3 during the protein isolation the high amount of Se containing amino acids is lost. And the digestion with the enzymes also results in the high lost of Se-containing amino acid component. Only ~7 % of the total organic Se is recovered when 5:1 ratio is used (Table 3). Is there any reference in which other isolation process (other or bigger fraction of CV protein was isolated) and enzyme(s) were used?

Reviewer 3 Report

Comments and Suggestions for Authors

This is a very very interesting study, reproting about the a selenium-rich food plant. Although the manuscript is written quite well, I suggest to adress the following aspects for getting more justification and recognition.

1) As it was submitted to Foods, it needs even more clearer realtion to food and in some parts food safety. When aiming at asupplements, they need to be save. Please find some words about that aspect.

2) However, I do not get the necessity of producing/yielding one peptide. Isn't it eough to just the proteins and these will be degraded in the gastrointestinal tract ?

3) In Europe, it could be that such peptides are regarded as novel food and need a legislative basis. Again, a few more words on global importance, necessity, and (food) safety would be good. Or this is just a Chinese thing ?

4) As indicated under point 1): Why is this food ? The way it is written, is more like a technological or bioeconomical study.The main focus is described to be on the immobilized enzymes. Consequently, journal selection wonders a little.

5) As this is a scientific journal, an objective scientific language should be used. Terms or words like "Regrettably," etc. are not scientific. They raise the feeling  that this also advertising a certain viewpoint.

6) In the same context, a scientific study should always provide a scientific hypothesis. Here, the scientific basis and justification for doing this are not really clear.

Comments on the Quality of English Language

This is OK.